# T-Cell Responses in Merkel Cell Carcinoma: Implications for Improved Immune Checkpoint Blockade and Other Therapeutic Options

**DOI:** 10.3390/ijms22168679

**Published:** 2021-08-12

**Authors:** Laura Gehrcken, Tatjana Sauerer, Niels Schaft, Jan Dörrie

**Affiliations:** 1Department of Dermatology, Universitätsklinikum Erlangen, Friedrich-Alexander-Universität Erlangen-Nürnberg, Hartmannstraße 14, 91052 Erlangen, Germany; Laura.Gehrcken@extern.uk-erlangen.de (L.G.); Tatjana.Sauerer@uk-erlangen.de (T.S.); Niels.Schaft@uk-erlangen.de (N.S.); 2Comprehensive Cancer Center Erlangen European Metropolitan Area of Nuremberg (CCC ER-EMN), Östliche Stadtmauerstraße 30, 91054 Erlangen, Germany; 3Deutsches Zentrum Immuntherapie (DZI), Ulmenweg 18, 91054 Erlangen, Germany

**Keywords:** immune checkpoint blockade, Merkel cell carcinoma, immunotherapy, T-cell response, large T antigen, tumour mutational burden, Merkel cell polyomavirus, tumour microenvironment

## Abstract

Merkel cell carcinoma (MCC) is a rare and aggressive skin cancer with rising incidence and high mortality. Approximately 80% of the cases are caused by the human Merkel cell polyomavirus, while the remaining 20% are induced by UV light leading to mutations. The standard treatment of metastatic MCC is the use of anti-PD-1/-PD-L1-immune checkpoint inhibitors (ICI) such as Pembrolizumab or Avelumab, which in comparison with conventional chemotherapy show better overall response rates and longer duration of responses in patients. Nevertheless, 50% of the patients do not respond or develop ICI-induced, immune-related adverse events (irAEs), due to diverse mechanisms, such as down-regulation of MHC complexes or the induction of anti-inflammatory cytokines. Other immunotherapeutic options such as cytokines and pro-inflammatory agents or the use of therapeutic vaccination offer great ameliorations to ICI. Cytotoxic T-cells play a major role in the effectiveness of ICI, and tumour-infiltrating CD8^+^ T-cells and their phenotype contribute to the clinical outcome. This literature review presents a summary of current and future checkpoint inhibitor therapies in MCC and demonstrates alternative therapeutic options. Moreover, the importance of T-cell responses and their beneficial role in MCC treatment is discussed.

## 1. Merkel Cell Carcinoma

### 1.1. Definition of Merkel Cell Carcinoma

Merkel cell carcinoma (MCC) is a rare but aggressive neuroendocrine skin malignancy. It was first described as a “trabecular carcinoma of the skin” in 1972 by Cyril Toker [1]. Patients are usually diagnosed at a median age of 75–80 years, and only 12% are younger than 60 years [2,3]. Often the primary tumour has already metastasised locoregionally or to the lymph nodes at the time point of diagnosis [4].

Due to its aggressiveness, the five-year survival rate is as low as 40% [5,6] and even worse for nodal (35%) or distant (14%) metastasis [7]. More than one third of the patients diagnosed with MCC dies from this disease [4]. The recurrence rate varies strongly from 26% [8,9] to 60% [10], depending on the stage and whether only local or regional recurrence was investigated. In the last couple of years, the incidence of MCC has been rising, not only in the USA (from 0.44 cases/100.000 in 2000 to 0.66 cases/100.000 in 2016) [3,11,12] but also in Sweden (from 0.09 cases/100.000 in 1993 to 0.2 cases/100.000 in 2012) [13] and Australia (from 1.6/100.000 in 1993 to 20.7/100.000 in 2010) [2]. Increasing numbers could be explained by better immunohistological staining, new diagnostic markers, and behavioural changes, for example due to increased sun exposure in Australia, which leads to more mutations.

In general, two types of MCC can be distinguished. One is caused by the Merkel cell polyomavirus (MCPyV) and the other one by chronic UV-light exposure. The majority of patients bears MCPyV^+^ tumours (80%) [14], for which the integration of the viral genome into the host genome is characteristic. The remaining 20% of cases are virus negative (MCPyV-negative) and associated with chronic UV-light exposure that causes a high mutational load, leading to MCC by mechanisms not yet fully understood [15,16].

The cell type of origin of the Merkel cell carcinoma is still discussed. Originally, it was thought that the cancer rises from Merkel cells due to similar morphological features. However, Merkel cells are found in the basal layer of the epidermis of the skin, while MCC cells are mostly detected in the dermal layer of the skin and additionally express different markers compared to Merkel cells. It is, therefore, now rather assumed that the origin of MCC are dermal cells. However, there seem to be different origins for both MCC types. Although it is thought that MCPyV-negative MCC is more often associated with dermal keratinocytes [17] or early progenitor cells, for MCPyV^+^ MCC it is likely that the virus targets dermal fibroblasts [18] and productively infects them. Liu et al. hypothesize that the virus enters Merkel cells “accidently” and causes MCC [18].

Merkel cell carcinoma regularly appears in anatomical areas that are highly exposed to sunlight, such as the neck or the face [19]. The clinical features for diagnosis are called **AEIOU** factors [20]. They describe an **A**symptomatic nodule that **E**xpands rapidly. It occurs under **I**mmunosuppression in people who are **O**lder than 50 with a location at **U**V-exposed sites. Around 89% of patients meet at least three of these criteria [20].

Some of the AEIOU factors are risk factors, including advanced age, UV-exposure and immunosuppression. It was shown that immune-compromised patients with lymphocytic leukaemia had a 30-fold increased chance of developing MCC compared to immune-competent patients [20]. Another example are HIV patients, whose risk of developing MCC is increased 13-fold compared to healthy individuals [21]. Paulson et al. showed that the 3-year survival rate in immune-competent patients is twice as high as in immune-compromised patients [22]. This suggests that a functional immune system plays a very important role in keeping the tumour under control. Advanced age, as another risk factor, leads to immune senescence, an age-related alteration of the immune system and a loss of T-cell receptor repertoire [23]. This could explain the increased incidence of MCC in elderly people.

Mutagenesis is a frequent event related to cancer development. Patients with MCPyV^+^ MCC show a low number (12.5 per-exome) of somatic single nucleotide variants (SSNVs) while MCPyV-negative MCC harbours around 1121 SSNVs per-exome [15]. The two most commonly mutated genes in MCPyV-negative tumours are tumour protein 53 (TP53), which encodes for p53, and retinoblastoma-associated protein 1 (RB1), both important regulators of the cell cycle and apoptosis [15]. Nevertheless, it is known that MCPyV-negative MCC is associated with a high tumour mutational burden (TMB), which could be useful for immunotherapy as more tumour neo-antigens can be used as targets [15]. The MCPyV-negative MCC also shows a typically UV-mutational signature. In comparison to the non-viral MCC, the viral MCC does not have this UV-mutational signature and a low TMB [24].

### 1.2. The Merkel Cell Polyomavirus

The Merkel cell polyomavirus (MCPyV) is a small, non-enveloped double-stranded DNA virus and the only known virus from the family of polyomaviruses that is thought to cause cancer in humans. The primary infection is usually asymptomatic with a high seroprevalence of about 70% in the human population [25], so that the virus has evolved to co-exist with its host. Usually, the healthy human immune system is able to keep the virus replication under control and the asymptomatic infection with MCPyV generally occurs during childhood but the seropositivity increases with age, from around 50% in children under the age of 15 years to 80% in people over 50 years [26].

The incidence of only 2000 new MCC cases every year in the US with a seroprevalence of 70% of MCPyV among the population raises the question why MCC is still a quite rare cancer type. Nevertheless, new treatment strategies for MCC are rapidly evolving. This is underlined by numerous recently published reviews concerning MCC and new trends in treatment [27,28,29]. The discrepancy between the ubiquity of the MCPyV and the rare manifestation of MCC is the consequence of the very improbable molecular events; it occurs by mistake in combination with several other factors resulting in viral transformation of the host cell.

In addition, the virus can be found in a lot of tissues in the body such as the skin, the saliva, or the aerodigestive tract, but only the neuroendocrine cells of the skin are susceptible to transformation by MCPyV [30].

### 1.3. The Molecular Steps in the Evolution of MCC

Prior to causing MCC by MCPyV, two separate events need to take place. First, the virus genome needs to be linearized and integrated into the host genome, and secondly the viral genome needs to acquire a specific mutation. Integration into the host genome occurs at random sites and by accident, for example via viral fragmentation during viral replication [31]. UV light can then cause mutations in the viral genome.

The virus genome consists of an early region (ER), and a late region (LR), as well as a non-coding control region. The LR encodes the viral capsid proteins VP1 (major) and VP2 (minor), both relevant for productive infection of new host cells. The ER encodes the two oncogenic proteins: the tumour (T) antigens, namely the large T antigen (LTA) and the small T antigen (STA). Both share 80 amino-terminal (N-terminal) amino acids [32].

The mutations lead to a C-terminally truncated form of the LTA, referred to in the following as truncLT if mutated. The full-length LTA has a carboxy-terminus (C-terminus) containing an origin binding domain and a helicase domain. Both are required for virus replication. Full-length LTA can also indirectly bind p53 and thus interfere with its activation [33]. Commonly p53 is known as a tumour suppressor, negatively regulating genes in the cell cycle. If the activation of p53 is suppressed, the capability to control the cell cycle and proliferation are lost.

Nonsense or frameshift mutations produce a premature stop codon that leads to the truncLT. Due to the C-terminal truncation, the helicase domain is lost and the virus is no longer able to replicate [34], therefore the T antigens are expressed in the host cells. Thereupon the only regions left are the Rb-binding domain and an N-terminal J region. Included in the Rb-binding domain is a conserved LXCXE Rb factor binding motif. The Rb protein plays a significant role in the cell cycle, controlling the progression towards S-Phase, inhibiting E2F, a transcription factor. By repression of E2F, it prevents the activation of genes important for the progression of the cell cycle. The truncLT binds with high affinity [33] and inhibits Rb. Therefore, the cells progress to S-Phase, which leads to increased proliferation [35].

In contrast to the LTA, the STA keeps its full length and is thought to play a role in cell survival, as well as proliferation of MCC cells [36], but its full function is still unknown. In MCPyV^+^ cell lines, silencing of the T antigens led to growth arrest or cell death [37], demonstrating their importance as oncogenes in the process of causing MCC. Like other intracellular proteins, the truncLT is processed in the cytoplasm and presented on major histocompatibility complex (MHC; also termed human leukocyte antigen, HLA in humans) class I to cytotoxic CD8^+^ T-cells, giving rise to diverse peptide epitopes, which can be recognised by T-cells and cause an immune response against the tumour.

### 1.4. Immunogenicity and Immune Escape in MCC

Developing malignancies are under constant immune surveillance and undergo a process termed immunoediting: the relation between tumour cells and the immune system changes [38]. At the beginning, in the so-called elimination phase, the immune system can keep the tumour growth under control. Over time the tumour will gain more mutations, allowing it to eventually evade the immune system. In the second phase, the equilibrium, the immune system keeps the balance. The tumour does not grow anymore, but is also not reduced in size. In the escape phase, the immune system cannot control the tumour growth anymore. As a result, the tumour escapes and grows by creating an immune suppressive environment.

During recent years, it was shown that the cellular immune response is especially important to fight MCC. The immunogenicity of MCC is either based on the presence of the viral T antigen in MCPyV^+^ MCC, or the high mutational burden in MCPyV-negative cases. A recent publication indicated that the viral status can be used as a prognostic marker, and that MCPyV^+^ tumours are associated with a better overall survival, as well as recurrence free survival [39]. T-cells, specifically cytotoxic CD8^+^ ones, play a very critical role. The presence of many intratumoural CD8^+^ T-cells is associated with better prognosis and survival [40].

However, not only T-cells play a role in the prognosis of MCC. Antibodies against MCPyV T antigens produced by B-cells correlate with disease burden and MCC recurrence [41]. Anti-MCPyV T antibody titres decrease after successful treatment and upon recurrence they increase again. For these reasons, anti-MCPyV T antibody titre is used to monitor patients after MCC treatment. Unfortunately, seropositivity for MCPyV cannot be used for MCC screening or to distinguish MCPyV^+^ from MCPyV-negative patients due to the high seropositivity levels in the healthy population.

As described above, the cancer evolves with strategies to suppress or evade the immune system, which conceal or protect the tumour cells from the immune cells. One example is the down-regulation of the expression of MHC I molecules on the surface of MCC cells [42]. Another mechanism is the creation of an immune suppressive tumour micro-environment with the help of suppressive cytokines and chemokines such as TGF-β or IL-10 [43,44,45]. Otherwise, an up-regulation of checkpoint proteins on the surface of T-cells, such as programmed cell death protein 1 (PD-1) [46] or cytotoxic T lymphocyte associated protein 4 (CTLA-4) [47], is detected. PD-L1 has been shown to be up-regulated in tumour cells and immune cells in the tumour microenvironment of MCC patients with a better overall survival, if PD-L1 is overexpressed [48,49]. This suggests that these checkpoints are a good target for immunotherapy using checkpoint inhibitors, which is discussed in the next section.

## 2. Checkpoint Inhibitors

### 2.1. The Success Story of Immune Checkpoint Inhibitors

Immunotherapies emerge as an increasingly important treatment option for cancer patients. In contrast to chemotherapy, which does not only kill tumour cells, but also healthy cells, immunotherapies can modulate the immune system of patients against the tumour. Nevertheless, there are currently several clinical trials testing chemotherapy in combination with checkpoint inhibitor therapy (see Table 1).

In recent years, immune checkpoint inhibitors (ICIs) became one of the most promising immunotherapeutic type of drugs. These antagonistic antibodies interfere with the “brakes” of the immune system. Those “brakes” usually regulate lymphocytes and prevent autoreactive cells from causing autoimmunity. This mechanism is essential in avoiding autoimmune diseases, but in cancer patients it affects the ability to fight off the cancer by down-regulating the immune response. As described above, the tumour uses this as one of multiple evasion strategies.

So far, around 50% of MCC patients benefit from the development of ICIs, and their use is now approved as standard of care for metastatic MCC (mMCC) [50]. Prior to the development of ICIs, chemotherapy was the main treatment. However, chemotherapy is not able to eradicate MCC completely, which is why the recurrence rates are high. Another issue is the development of chemoresistance and short durable responses with less than 8 months after treatment [51,52]. Furthermore, the complete response rates decrease with line of treatment. Although the first-line treatment response rates range from 20–60%, they become worse for second-line (23–45%) treatments [52]. Additional therapy alternatives such as radiotherapy were recently reviewed in Zwijnenburg et al. [53] and Babadzhanov et al. [54]. New treatment options and novel agents are needed to induce durable and effective responses in MCC patients.

The first immune checkpoint was discovered in 1987 by Brunet et al. and was named CTLA-4 [55]. Several years later, in 1995 Krummel et al. and Leach et al. showed that CTLA-4 negatively regulates T-cell activation and concluded that inhibitory antibodies against CTLA-4 can be used to enhance anti-tumour immunity [56,57]. Ipilimumab, an anti-CTLA-4 antibody, was the first checkpoint inhibitor approved by the Food and Drug Administration (FDA) in 2011 for the treatment in metastatic melanoma. In the following years, ICIs were extended to the PD-1/PD-L1 axis, and more antibodies were approved, for instance Pembrolizumab in 2014 as the first anti-PD-1 antibody.

Since T-cells often display an exhausted phenotype in different cancers including MCC [47,58], clinical trials were performed to test the efficacy of the ICIs in MCC. The exhausted phenotype is, among other exhaustion markers, characterized by the presence of PD-1 on the surface of T-cells, while the tumour cells and immune cells in the tumour microenvironment (TME) display PD-L1 on their surface, making MCC susceptible for ICI treatment.

Immune checkpoint receptors can be expressed on all lymphocytes such as T- and B-cells. The major inhibitory checkpoint-ligand combinations are B7.1 (CD80)/CTLA-4 or B7.2 (CD86)/CTLA-4, and PD-L1/PD-1. Both checkpoints, CTLA-4 and PD-1 try to confine the T-cell response within a physiological range and protect the host from autoimmunity.

As the ligands of CTLA-4 are only expressed on professional antigen-presenting cells (APCs), it acts mainly on T-cell activation in peripheral lymphatic tissue, where it competes with the co-stimulatory receptor CD28 for the binding to B7. Upon activation of a T-cell, CTLA-4 expression is up-regulated and therefore thought to be a negative feedback loop that prevents over-activation of the T-cell. PD-1 in contrast is involved in prevention of tissue autoimmunity, although some professional APCs such as dendritic cells (DCs) can also express the corresponding ligand PD-L1. PD-1 appears on chronically stimulated T-cells and causes the inhibition of the T-cell function via downstream signalling.

In the TME, which will be discussed below, the expression of checkpoint molecules increases, mediating local immune resistance and evading the immune response. Ligands binding to the checkpoint receptors are up-regulated on the tumour cells or on APCs in the TME.

ICIs intent to reverse this blockade of the T-cells by either binding to CTLA-4 (Ipilimumab), PD-1 (Pembrolizumab, Nivolumab) or PD-L1 (Avelumab), unblocking an anti-tumour response against MCC.

### 2.2. Checkpoints Inhibitors in the Treatment of MCC

**Pembrolizumab** (*Keytruda*, MK-3475, Lambrolizumab) is an IgG4 anti-PD-1 humanised monoclonal antibody. With the first 26 patients from a Phase II multicentre non-controlled study from 2016 (KEYNOTE-017, NCT02267603, see Table 1), Nghiem et al. could show that MCC patients without any previous treatment, receiving 2 mg/kg Pembrolizumab every three weeks, had an objective response rate of 56%. The progression free-survival rate at 6 months was 67%. However, 15% of patients showed grade 3 or 4 adverse events (myocarditis and elevated levels of aminotransferase). In general, the trial showed that Pembrolizumab as a first-line treatment in MCC patients is working and is only associated with adverse side effects in some patients [59].

When extending the analysis to 50 patients and a median follow-up time of 14.9 months (KEYNOTE-017, NCT02267603, see Table 1) Nghiem et al. showed tumour control in both MCPyV^+^ and MCPyV-negative tumours. The clinical outcome was better, compared to patients who only received chemotherapy as a first-line treatment [60]. Based on the data from this trial, the FDA approved Pembrolizumab in 2018 for the treatment of recurrent locally advanced or metastatic Merkel cell carcinoma [61].

**Avelumab**, (*Bavencio*, MSB0010718C), a human IgG1 monoclonal antibody, targets the ligand PD-L1. In 2017, the FDA approved Avelumab as the first checkpoint inhibitor for the treatment of mMCC. This approval was based on the results of the following trial.

This phase II, multicentre, and single-arm trial (JAVELIN Merkel 200 trial, part A; NCT02155647, see Table 1) by Kaufman and colleagues assessed the efficacy of Avelumab in patients with metastatic refractory MCC who already had undergone different unsuccessful treatment options such as chemotherapy. Of the 88 enrolled patients, about 40% were six-month progression free and the six-month durable response rate was around 26% [62]. An update on these patients was published in 2018 after a 12-month follow-up period. MCC patients that had been treated with Avelumab a year earlier showed an objective response rate of 33%, as well as a durable response in general and promising survival outcomes [63]. A follow-up analysis over 36 months after treatment with Avelumab showed no additional adverse effects and the response was ongoing for over 40.5 months after treatment [64].

In the JAVELIN Merkel 200 trial, part B (NCT02155647, see Table 1), performed by D’Angelo et al. in 2018, the efficacy and safety of Avelumab as first-line treatment was evaluated. The pre-planned interim analysis of the phase II trial of 39 patients with mMCC showed an overall response rate of 62.1% but also grade 3 related adverse events [65].

Confirming the results from these trials, an observational study with patients treated for advanced MCC was performed in the Netherlands. For the 40 patients receiving the treatment, the relative response rate was 50%, whereby 28% of patients demonstrated a complete response (CR). CR was reached in patients receiving Avelumab as a first-line treatment, showing the importance of using ICIs as a first-line treatment to be effective. After all, 14% of their patients responded with CRs to second-line treatment [66].

Drusbosky et al. performed a case study in 2020 combining Avelumab treatment with the interleukin (IL)-15 agonist N-803 in combination with Abraxane^®^ in a patient with advanced mMCC. This patient did not show durable responses to chemotherapy, surgery, and Avelumab monotherapy before [67]. IL-15 plays a role in controlling survival and turnover of memory T-cells. A study in MCC cell lines showed that IL-15 alone enhanced cytokine production. Together with IL-2 it induced even antigen-independent proliferation of T-cells [47]. Abraxane^®^, which is paclitaxel bound to albumin, belongs to the group taxanes that are often used in chemotherapy. Taxanes induce cellular arrest and lead to activation of macrophages, therefore initiating an anti-tumour response [68]. The combined treatment consisting of an ICI, a chemotherapeutic drug, and an interleukin lead to a complete response in this patient suggesting it as a promising treatment combination for mMCC [67].

As discussed in Section 1.4, increased PD-L1 expression is associated with better survival compared to PD-L1 negative patients and PD-L1 expression is mostly associated with the presence of the MCPyV [48,69]. Therefore, the question arises, if anti-PD-L1 is efficient as first-line of treatment for these patients. Several studies showed that MCC patients respond to the anti-PD-L1 antibody Avelumab independently of their PD-L1 or viral status, suggesting that different underlying mechanism cause the therapeutic benefit [62,65].

**Nivolumab** (*Opdivo*, MDX1106) is a humanised IgG4 anti-PD-1 monoclonal antibody. Until now, it is not approved for treatment of MCC by the FDA, but for other cancers such as Hodgkin’s lymphoma [70], advanced melanoma [71], or metastasised small cell lung cancer [72].

A case report from Walocko et al. showed a durable response towards Nivolumab monotherapy treatment in a patient presenting with mMCC [73]. Nivolumab monotherapy is currently under investigation in the ADMEC-O phase II trial (NCT02196961, see Table 1). In the CheckMate358 Trial (NCT02488759, see Table 1) performed by Topalian et al., the efficacy and safety of Nivolumab in the neo-adjuvant setting was investigated. Administration occurred 4 weeks before surgery on patients with a stage III resectable MCC. Thirty-six of the patients who underwent surgery and had the pre-surgical Nivolumab treatment showed a response rate of 50%. Additionally, they showed that the efficacy of the anti-PD-1 antibody did not correlate with baseline MCC viral status or PD-L1 expression. In summary, the administration of Nivolumab in a neo-adjuvant setting together with surgery was generally safe for the patients and led to a pathological complete response (pCR) in 47.2% of the patients [74].

The human monoclonal anti-CTLA-4 IgG1 antibody **Ipilimumab** (*Yervoy*, MDX-010) was, as described above, FDA approved in 2011 for the treatment of metastatic melanoma [75]. Due to the development of PD-1/PD-L1 ICIs, Ipilimumab is not used as a monotherapy in MCC. In addition, a multicentre study, investigating Ipilimumab monotherapy in MCC patients, showed disease progression despite treatment and increased toxicity [76].

Compared to monotherapy, the combination of a CTLA-4 and a PD-1/PD-L1 blocking ICI can improve the therapeutic effect for melanoma or advanced renal cell carcinoma [77,78]. Nevertheless, using two ICIs at once could increase the incidence of treatment-related adverse events as seen in melanoma patients treated with Ipilimumab and Nivolumab compared to patients treated only with Ipilimumab or Nivolumab [79].

In a case report from 2019, one patient with stage III MCC, resistant to Avelumab monotherapy, was treated with a combination of Nivolumab and Ipilimumab (dose: Ipi 3 mg/kg + Nivo 1 mg/kg). Four weeks after receiving the treatment, he showed complete remission [80]. In the same year, another report was published, showing similar results in a retrospectively analysed small cohort of 13 patients (dose: Ipi 1 mg/kg + Nivo 3 mg/kg). This suggests that combined therapy of ICIs is a possible second-line treatment option for patients with refractory anti-PD-1 MCC [81]. A comparable case study with 4 patients was reported from Germany in 2020, showing an overall response rate of 60% to combination treatment of Nivolumab and Ipilimumab (dose: Ipi 1 mg/kg + Nivo 3 mg/kg) in melanoma [82]. A case study performed by Glutsch et al. in Avelumab-refractory MCC patients treated with Nivolumab and Ipilimumab (dose: one half of the patients was treated with 1 mg/kg Ipi + 3 mg/kg Nivo or 3 mg/kg Ipi + 1 mg/kg Nivo) showed no grade 3 or 4 immune-related events and a good response towards treatment [83]. Taken together, the combination of different ICIs improved the outcome of patients with a primary resistance to one ICI, and therefore their disease outcome.

Overall, these different checkpoint inhibitors show that about half of the patients with MCC show increased survival when being treated with them in a first-line setting (summarized in Table 2). Nevertheless, some shortcomings with the current ICIs are discussed in the next section and it is important to investigate other therapeutic approaches as well.

### 2.3. The Limits of ICI

Even though the response rates of ICIs are quite good in MCC, still 50% of patients do not respond to the treatment due to possible resistances which can be either primary, i.e., from the beginning of the treatment or secondary, i.e., when a tumour first responds but later the treatment stops working and the patients relapse. Reasons for resistance are manifold and all prevent an effective interaction of T-cells and tumour cells: if a patient has a priori no MCC-specific T-cells, a release of the inhibitory breaks cannot create a T-cell activity against the tumour [84]. This lack of T-cells can be caused by absence of suitable tumour antigens or by an absence of corresponding TCRs in the patient’s repertoire. However, even if proper antigens are expressed in the tumour, they need to be presented on MHC molecules, and if the cancer cells’ antigen presentation is incapacitated, e.g., by down-regulation of MHC [42,85] they will be invisible for the T-cells. In addition, the tumour microenvironment can prevent the T-cells from entering the tumour or act heavily immunosuppressive, so entering T-cells are rendered anergic or display an exhausted phenotype, which cannot be reverted by ICI treatment.

As written above, ICI treatment seems to be the most beneficial if applied in a first-line setting rather than as second-line treatment after e.g., an unsuccessful chemotherapy. The combination of two ICIs showed some promising results, but Knepper et al. analysed genomic data of 317 MCC tumours and found that patients with a resistance to one ICI do not benefit from just switching to another ICI. Instead, they suggest offering those patients an alternative immunotherapy by taking part in a clinical trial [24].

Consequently, it is important to overcome possible resistances and find treatment options that synergize with ICI. The next section will explore such strategies further.

### 2.4. Extended Therapeutic Options for MCC Patients

Treatment with ICI has been the greatest advance in treatment of MCC so far. Nevertheless, a substantial part of the patients, especially those with metastatic disease, do not benefit from the treatment. Hence, a variety of alternative treatment strategies was explored, which may be—or are already—combined with ICI.

#### 2.4.1. Cell Growth Inhibitors

If ICI alone cannot interfere with the growth of the tumour in 50% of the cases, it could be beneficial to combine ICIs and certain chemicals that inhibit cell growth. A phase II trial investigates the efficacy and safety of Avelumab treatment together with Domatinostat (NCT04393753, see Table 1) for MCC progressing after anti-PD-1/PD-L1 therapy. Domatinostat is a histone deacetylase inhibitor, which leads to G2/M cell cycle arrest and initiates apoptosis in MCC cell lines showed by Song et al. Additionally, they observed an up-regulation of MHC I on the surface, which might result in higher presentation of neo-antigens on MHC and therefore a better anti-tumour T-cell response. [86].

#### 2.4.2. Cytokines and Toll-Like-Receptor Agonists

Considering intratumoural T-cells as crucial players in fighting of MCC [40], another therapeutic option is to increase T-cell numbers and to strengthen their response with the help of cytokines or toll-like-receptor (TLR) agonists.

Cytokines as interleukins (IL) can be pro-inflammatory and beneficial in a tumour setting. Bhatia et al. published data from a trial testing an intratumoural application of an IL-12-encoding plasmid via in vivo electroporation in mMCC patients. Their data showed a durable response in a subset of patients, without severe adverse effects, and an overall response rate of 25% in tumour-bearing patients. They further demonstrated that the intratumoural expression of IL-12 [87] induced the MCPyV-specific T-cell responses, and in comparison to systematic application of interleukin, local application prevents systemic toxic effects. Hence the combination with ICI treatment seems very promising.

TLRs are a crucial part of the innate immune system, being present on the surface of cells recognising for example lipopolysaccharides (LPS), or being inside the cell in endosomes recognising RNA and DNA, and inducing an interferon (IFN)-based immune response. Stimulation of these TLRs with artificial agonists could enhance the anti-tumour response.

Bhatia et al. performed another study where they used G100 (Glucopyranosyl lipid A), a TLR4 agonist, to overcome immune suppressive mechanisms (NCT02035657) [88]. TLR4 is present on several immune cells, such as DCs and macrophages, where it recognises LPS and induces an inflammatory response, which facilitates T-cell activation. Additionally, it directs T-cell differentiation into the direction of a Th1 response. Such a Th1 response is thought to be beneficial for tumour rejection by inducing cellular cytotoxicity and recruiting innate immune cells such as macrophages. In this trial G100 was tested in combination with surgery. G100 was able to reverse the immune suppression and re-establish an active immune response. This suggests that G100 can be used as a co-treatment together with anti-PD-1/-PD-L1 antibodies [88]. Other clinical trials testing cytokines and toll-like-receptor agonists are currently running (see Table 1).

#### 2.4.3. Nitric Oxide Blockers to Improve Extravasation

Extravasation is important for T-cells to reach the site of infection or the tumour site, and crucial mediators of this process are E-selectins. A down-regulation of E-selectin in the vasculature results in a decreased T-cell recruitment from the blood. Afanasiev et al. found that E-selectin is down-regulated by nitrogen oxide (NO) in the majority of MCC cases, correlating with poor T-cell infiltration. Higher E-selectin expression, in contrast, was associated with better survival [89]. Therefore, the combination of NO blockers and ICIs could cause more T-cells to enter the tumour, where they in turn are not inhibited by PD-1-stimulation, thus improving the treatment of MCC.

#### 2.4.4. Therapeutic Vaccines

Therapeutic vaccines are another treatment option. They can support tumour-specific T-cell development and promote anti-tumour immunity. The target antigen can be either cancer associated, i.e., germline-encoded but more or less specifically expressed in the tumour, or it can be cancer-specific, meaning that the antigen is absent from the host genome and solemnly expressed in the tumour cells. Since the latter are limited to the tumour tissue and hence not subject to central tolerance, they represent a better target compared to germline-encoded proteins. In MCC, the truncLT is such a cancer-specific antigen, which can be processed and presented on MHC molecules to T-cells, and therefore can induce an MCC-specific immune response. Since truncLT is important for the cancer to survive [37], the risk of antigen loss is low. Another advantage of truncLT is its similarity across different patients, allowing it to be used in a general application. However, the immunogenicity of truncLT is moderate [90,91], which is why the immunogenicity needs to be increased by using highly active DCs.

There are different methods how to deliver the antigens into the cells (reviewed in Tabachnick-Cherny et al., 2020 [92]). One possibility is the use of DCs as a therapeutic vaccine to induce and activate tumour-specific T-cells. Since DCs are primarily responsible to initiate adaptive T-cell responses, this may be a promising approach. For this purpose, usually monocytes from the patient’s blood are isolated and differentiated ex vivo into mature DCs. These DCs are manipulated to present the tumour antigens on the MHC molecules to the T-cells. They are subsequently transferred back into the patient, where they should initiate an anti-tumour response [93,94]. Common methods to load DCs with the corresponding antigens are pulsing with peptide or protein, or via mRNA-transfection [95]. This method is quite promising as it is highly individualized for each patient but also associated with high financial effort.

Our group showed in a preclinical evaluation that optimised DCs, activated by NF-κB and expressing the oncogenic driver of MCC, truncLT, result in a MCPyV-specific T-cell response [90]. The NF-κB-activated DCs were specially designed to produce IL-12 and induce memory-like cytotoxic T-cells. When electroporated with truncLT they were able to induce truncLT-specific T-cells from the blood of healthy donors and MCC patients. mRNA-electroporation of DCs allows the transient expression of the truncLT, which prevents the integration into the host DNA, avoiding the theoretical threat of a malignant transformation of the vaccine cells. Gerer et al. suggest a combination of ICIs and this method to increase the anti-tumour response [90].

To avoid the laborious process of ex vivo generation of dendritic cells, cell-free RNA formulations can be injected directly. Such mRNA vaccines showed great effectivity as preventive vaccines against COVID-19 and also represent a great potential for therapeutic cancer vaccination (reviewed by McNamara et al. [96] and Pardi et al. [97]). The first trial of that kind was reported by Weide et al. investigating the response to an mRNA vaccine, encoding for different melanoma specific antigens injected in 21 metastatic melanoma patients (NCT01684241). They showed a decrease in regulatory T-cells and an increase in vaccine-directed T-cells [98]. A more recent phase I clinical trial (NCT02410733) tested a liposomal RNA vaccine with tumour-associated antigens for advanced melanoma patients and an interim analysis revealed that the vaccine mediated durable objective responses in ICI-experienced patients [99]. So far, mRNA vaccines have not been tested in MCC patients, but due to observations with other kinds of cancer, they present another possible immunotherapeutic treatment option.

#### 2.4.5. Adoptive T-Cell Transfer

Another approach to increase the number of tumour-specific T-cells is adoptive T-cell transfer. One source for such cells are peripheral blood mononuclear cells (PBMCs) from the patient or healthy donors, which are activated ex vivo with a specific antigen, thus generating T-cells with high-affinity antigen receptors to overcome immune tolerance. Chapius et al. used MCPyV-specific T-cells generated ex vivo from the patient’s PBMCs together intralesionally applied with IFNβ to treat an MCC patient. They observed lasting accumulation of the transferred CD8^+^ T-cells in the lesions and a regression of 2 out of 3 metastases [100].

Davies et al. used monocyte-derived DCs from healthy donors to generate MCPyV-specific T-cells with the help of overlapping 15-mer peptide libraries of the MCPyV LTA and STA. This strategy generated mainly TA-specific CD4^+^ T-cells and suggest that this treatment option, together with other immunotherapies, could improve the outcome of MCC patients [101].

All the above-presented therapeutic options for ICI-resistant MCC patients offer new possible treatment strategies. Currently several clinical trials are ongoing to expand our knowledge further (summarized in Table 1). In addition, these therapeutic options underline the importance of T-cells in the fight against MCC, and therefore this review will have a closer look at T-cells in the next section.

## 3. The Importance of T-Cell Immunity in MCC

MCC tumours are highly immunogenic and the cellular immune response plays a crucial role in defeating the cancer. In addition to a lot of other functions, T-cells can recognise and eliminate the tumour cells. There are two basic lines of evidence that show how important a functioning T-cell response in MCC is: (1) CD8^+^ T-cell infiltration is associated with a better outcome [40], and (2) immune suppression is a strong risk factor [22].

### 3.1. The Role of Different T-Cell Subtypes in MCC

With their ability to kill tumour cells, CD8^+^ T-cells play the most important role and are in focus for the development of therapeutic approaches. The number of cytotoxic CD8^+^ T-cells in MCC and other cancer types was analysed in a study performed by Blessin and colleagues with the help of tissue microarrays. From data of 42 MCC samples they calculated a mean number of 156 CD8^+^ T-cells/mm^2^ compared to Hodgkin’s lymphoma where they found 1573 CD8^+^ T-cells/mm^2^, or only 6 CD8^+^ T-cells/mm^2^ in pleomorphic adenoma [102]. This indicates that MCC shows an intermediate level of CD8^+^ T-cell infiltration.

Paulson et al. reported that the major infiltrating subtypes found in MCC tumours are T-cells which highly express CD8β and CD3ε on their surface [40]. Both molecules are part of the CD3-T-cell-receptor (TCR) complex and play a crucial role in the recognition of antigens and activation of the T-cell. Part of the CD3-TCR complex is the TCR heterodimer consisting of an α and a β chain, which are connected by disulphide bonds. The TCR is responsible for antigen recognition, i.e., it recognises antigen peptides presented in MHC molecules. The CD8β chain interacts with the α3 domain of the MHC I molecule and is therefore important for the stabilisation of the antigen recognition. The CD3 complex consists of a CD3εδ and a CD3εγ heterodimer as well as a CD3ζζ homodimer. These subunits, harbouring intracellular immune receptor tyrosine-based activation motifs (ITAMs) allow signalling and lead to activation of the T-cell upon antigen recognition. The TCR and the multi-subunit CD3 complex are non-covalently associated (reviewed in Wucherpfennig et al., 2010 [103]). Therefore, an overexpression of both molecules, CD8β and CD3ε, could permit better antigen recognition and a faster activation of the T-cell. The favourable CD8^+^ T-cell infiltration into the tumour was only shown in 20% of MCC patients investigated by Paulson et al. [40], which suggests that immunosuppressive mechanisms prevent infiltration. However, even if the T-cells can infiltrate the tumour, they often display an exhausted phenotype or are dysfunctional.

The number of CD8^+^ T-cells specific for MCPyV in the blood of patients correlates with disease burden [46]. In MCPyV^+^ MCC, LTA and STA are the dominant trigger to activate the adaptive immune system, while in MCPyV-negative tumours the cause is the high TMB. Nevertheless, the CD8^+^ T-cells often display a reversible exhausted phenotype by expressing PD-1, T-cell immunoglobulin and mucin domain-containing 3 (TIM-3) and CTLA-4 on their surface due to chronic antigen exposure [46]. In that state the T-cells are not able to produce cytokines, proliferate or lyse other cells. Exhausted T-cells can be reactivated by ICIs that bind to those inhibitory receptors as mentioned earlier.

Not only CD8^+^ T-cells play a role in the fight against MCC. CD4^+^ T-cells have been shown to play a crucial role as well. The majority of CD4^+^ T-cells act as T helper cells and support the activation of CD8^+^ T-cells but also activate B-cells so they differentiate into antibody-producing plasma cells or memory B-cells. In healthy individuals, there is a high prevalence of antibodies specific for the MCPyV, which suggests a major role of virus-specific T helper cells [104]. In melanoma it was already shown that enhanced CD4^+^ T-cell responses contributed to increased anti-tumour CD8^+^ T-cell responses [105]. This leads to the conclusion that CD4^+^ targeted therapies are a possible co-treatment option in MCC to enhance CD8^+^ T-cell responses.

Another way to classify T-cells, apart from CD4 or CD8 expression confers to their development phase. Spassova et al. showed that the prevalence of central memory T-cells (T_cm_) among tumour-infiltrating lymphocytes (TILs) was higher in MCC patients responding to ICI treatment compared to non-responders [84]. These T_cm_ also displayed a highly diverse TCR repertoire among the TILs. TILs of non-responders, in contrast, predominantly showed the phenotype of terminally differentiated effectors and a constrained TCR repertoire [84]. It is thought that a higher and more diverse TCR repertoire can recognise more antigens compared to a low TCR repertoire. If there are more T-cells recognising different antigens, ICIs can reactivate more T-cells and the treatment can be more effective. Additionally, the low TCR repertoire was associated with terminal differentiation and an irreversible T-cell dysfunction, so these T-cells cannot be reactivated by checkpoint inhibitor therapy which could explain why some patients do not respond to the treatment [84]. The loss of TCR diversity is also seen among elderly people who have an increased risk of developing MCC [23]. This highlights the importance of a broad range of TCRs for a functional immune system and an effective anti-tumour response.

Tumours of MCC patients can continue to grow even if different effector T-cells are found in their proximity. This suggests the presence of other regulators in the tumour microenvironment (TME), leading to suppression of the immune response. An additional T-cell subtype found in the environment of the tumour are regulatory T-cells (T_regs_) [106], which suppress the immune response by secreting anti-inflammatory cytokines such as IL-10 or TGFβ, and are characterized by constitutive expression of the transcription factor FoxP3 [47]. Two inhibitory receptors on the T_reg_, the lymphocyte activation gene 3 (LAG3) and the T-cell immunoreceptor with Ig and ITIM domain (TIGIT) can interact with DCs and suppress their function. TIGIT can interact with poliovirus receptor (PVR, CD155) on DCs to induce IL-10 secretion for suppressing the Th1 T-cell response, inhibiting the NF-κB pathway and preventing the effective cytotoxic immune response, which is required to reject the tumour [107]. Chauvin et al. showed that IL-15 immunotherapy combined with a blockade of TIGIT leads to an increase in intratumoural NK cells in vivo and in vitro for MHC class I deficient melanoma [108]. Eventually a combinatorial immunotherapy of both drugs could also be tested for MCC tumours to see if this mediated an increase in tumour-specific NK cells as well. LAG3 binds to MHC II on DCs, therefore inhibiting the interaction of a DC with a CD4^+^ T-cell, and preventing their activation and maturation [109]. In addition to T_reg_, a variety of other cell types and immunosuppressive mechanisms can be active in the TME that give infiltrating T-cells a hard time, as discussed in the next section.

### 3.2. The Tumour Microenvironment in MCC

The TME consists of different cell types such as fibroblasts, dendritic cells, lymphocytes as well as the extracellular matrix, and blood vessels [110], and is mostly immunosuppressive. Different studies show that T-cells can be found in the TME, but their activation is suppressed by different mechanisms: (1) down-regulation of MHC complexes on APCs [42], (2) expression of inhibitory checkpoints [48], (3) T-cell exhaustion [47], (4) immunosuppressive cytokines such as IL-10 [111], and (5) regulatory T-cells [47].

Next to T-cells, another cell type found in the TME are CD68^+^ CD163^+^ tumour-associated macrophages (TAMs) [106,112]. These macrophages display an immunosuppressive M2 phenotype (reviewed by Sica et al., 2008 [113]). Just as with T_regs_, they secrete high levels of IL-10 and show a characteristic high PD-L1 expression on their surface [47,48]. Additionally, high levels of CD200 are found in the TME of MCC. CD200 inhibits classical macrophage activation and forces the precursors to differentiate into an M2 state. In addition, CD200 is also associated with a high level of T_reg_ distribution in the immunosuppressive environment. It was shown that an anti-CD200 antibody can effectively target CD200 in MCC in vivo, suggesting a possible treatment option for non-responders. However, until now no correlation could be found between the presence of CD68^+^ CD163^+^ TAMs and the overall survival [114].

Another cell type is a heterogeneous group of myeloid-derived suppressor cells (MDSC) in the TME [69]. These cells are mostly positive for CD33 and have a high PD-L1 expression, and consist of granulocytes, macrophages, and dendritic cells. They inhibit effector cell functions and promote tumour growth. Mitteldorf et al. described that tumour-cell conglomerates are surrounded by a barrier of PD-L1-expressing immune cells, which act as a “gatekeeper” preventing the T-cells from entering the tumour [69].

Both MDSC and TAMs are cell types that closely interact with one another and have anti-inflammatory and pro-tumoural characteristics. MDSCs drive the differentiation of macrophages into the immunosuppressive M2 type, which leads to a decreased production of pro-inflammatory IL-12 and a Th2 response. Macrophages are per se able to present antigens on MHC molecules, but in the tumour environment, MDSCs lead to a down-regulation of MHC II on the TAMs via IL-10 [115]. In response to the IL-10, secreted by the MDSCs, low amounts of IL-12 and high amounts of IL-10 are produced by the TAMs. The IL-6 produced by the TAMs leads to an up-regulation of IL-10 in MDSCs, which then influence the TAMs. These indirect effects of the crosstalk between both cell types influence the tumour microenvironment. Direct effects of the crosstalk between MDSCs and TAMs include T-cell proliferation arrest, inhibition of T-cell signalling via NO, induction of T_regs_, promotion of angiogenesis or down-regulation of CD3ε, which leads to less cytokine production (reviewed by Ugel et al., 2015 [45]). The MDSC-TAM crosstalk is a regulator of the homeostatic balance in the tumour environment.

Kervavrec et al. reported an improved survival in patients showing intratumoural CD8^+^ T-cells and CD33^+^ myeloid cells. Due to that finding, they suggest that these CD33^+^ cells cannot be MDSCs, otherwise the survival would be decreased. They also found them in non-necrotic areas with expression of MHC molecules and without immunosuppressive function [112].

A subset of MCC patients presents a low expression of cutaneous lymphocyte-associated antigen (CLA) on T-cells in the TME [47]. CLA is a specific homing receptor for the skin. Inside the skin-draining lymph nodes, the T-cells encounter tumour antigens presented by APCs. After activation of the T-cells, CLA is up-regulated and the cell is directed towards the skin, where the antigen originated from. In MCC, CLA seems to be down-regulated, suggesting that T-cell homing to the skin is impaired and therefore not enough T-cells reach the tumour. However, this study could not detect a correlation between CLA^+^ T-cells and a better survival in MCC patients [47].

In addition, those T-cells expressed low levels of CD69 and CD25 [47]. As markers of early (CD69) and later (CD25) T-cell activation, low expression of both markers further confirms that in the TME of MCC, T-cells have a less activated phenotype.

In addition to the mentioned cell types there are also natural killer (NK) cells in the TME [106]. NK cells recognise cells with down-regulated MHC complexes, because this negative signal triggers NK-cell activation and the release of its granule contents. It is suggested that NK inhibitory receptors such as killer-cell immunoglobulin-like receptors and NKG2A are up-regulated in the TME and prevent immune stimulatory secretion of IFNγ or TNFα, which are important for T-cell activation [116]. A study on a series of cases with 23 MCPyV^+^ MCC patients showed an improved survival of patients with a moderate or high number of tumours infiltrating NK cells, suggesting NK-cell infiltration as a prognostic marker [117].

A further relevant development in the tumour is the formation of tertiary lymphoid structures (TLS), which presence in the TME of MCPyV^+^ MCC correlated with recurrence free survival and better prognosis [118]. TLS are structures that contain T-cell zone- and B-cell-rich follicle-like areas [119]. Studies have demonstrated that the presence of B-cells and TLS play a role for the response to checkpoint blockade for example in melanoma [120,121] or HPV-associated squamous cell carcinoma [122]. Additionally, HIV infection is known to interfere negatively with germinal centre B-cells and T follicular helper cells (T_fh_) via PD-1/PD-L1 interference [123]. T_fh_ interact closely with B-cells in germinal centres to induce antibody production. Alonso et al. showed in a lung cancer mouse model that anti-PD-1 therapy increases the number of circulating T_fh_ and therefore additionally increases the number of TLS, leading to impaired tumour growth and support T_fh_-associated antibody production [124]. In contrast to the above-mentioned cases, Miller et al. showed that under PD-1 blockade, B-cell antibody production is not increased by checkpoint inhibitor treatment. Increased oncoprotein antibody production was rather observed in patients not responding to ICI treatment at all [125]. Therefore, the role of B-cells in MCC and the response to ICI treatment is an ongoing discussion. Nevertheless, it would be worthwhile to investigate the role of TLS in MCC in the context of ICI treatment.

Altogether, the TME (summarized in Figure 1) plays a crucial role in controlling the immune response against the tumour, and a further increased understanding will help to design new combinational treatment options.

### 3.3. The Influence of the MHC Complex in MCC

As discussed earlier, one major problem with MCC and many other cancers is immune evasion. It is thought that in the majority of MCC patients, who do not respond to ICI treatment, the tumour cells have downregulated the MHC molecules on their surface [42,85,126]. If antigens cannot be presented to the T cells, a release of the brake inhibiting the T cells is not resulting in a stronger immune response. For the specific anti-tumour response MHC molecules are essential, as they present the tumour-neo-antigens to T cells. The peptides presented on MHC I to CD8^+^ T cells generally have a specific length of 8-10 amino acids, while the MHC II complexes display antigens of various length with more than 13 amino acids to CD4^+^ T cells. This allows both types of T cells to get activated by a broad variety of tumour antigens. It is therefore important to re-induce the MHC molecules if they are absent. 

Since it is known that MHC molecules are often downregulated in MCC, Ritter et al. showed that epigenetic priming can restore MHC I molecules on the surface of APCs in MCC patients by inhibiting histone deacetylases (HDAC). They propose epigenetic silencing of the antigen processing machinery (APM) via histone hypoacetylation as the reason for downregulation [85]. HDAC-inhibitors have already been tested in MCC cell lines and showed a strong induction of MHC I expression [86]. Khan et al. showed that HDAC-inhibitor treatment activated proteasomal components like TAP and LPMP2, which led to an increase in proteasomal mRNA and a higher expression of MHC complexes on the cell surface [127]. 

In a case series performed by Ugurel et al. with four patients suffering from mMCC after PD-1/PD-L1 immunotherapy, two of them were treated with Nivolumab in combination with Panobinostat. Panobinostat is a HDAC inhibitor and already approved for the treatment of multiple myeloma in Germany. In one patient they detected a restoration of the expression of APM-associated genes and enhanced infiltration of CD8^+^ T-cells, 8 weeks after treatment. Unfortunately, the patients did not show a clinical benefit from the treatment. They suggest that this is due to the late onset of combination therapy. However, the treatment was well tolerated but needs further investigation in clinical trials [126]. In conclusion, HDACs seem to be a pharmaceutic target to increase the MHC expression, as well as inhibiting the tumour-cell growth as mentioned earlier.

### 3.4. The Role of Cytokines in T-Cell Responses in MCC

Cytokines are important modulators of the immune response and determine the development of T-cells. As stimulatory or modulatory molecules they regulate the differentiation in T-cell development and therefore the T-cell subtype. One important group of cytokines are interferons (IFNs), which play a major role in initiating mostly anti-viral but also anti-tumour responses. IFNs can be subdivided into three main classes: type I IFN (α, β, ε, κ, ν, ω, τ, δ, ζ), type II IFN (γ) and type III IFN (λ). While type I and III IFN are mostly associated with anti-viral responses, type II IFN is associated with activation of T-cells and NK cells.

In MCC cell lines it was shown that type I IFN treatment (with IFNα and IFNβ) impaired proliferation, metabolism, and viability of the tumour cells. Nevertheless, they also inhibited the expression of LTA and induced pro-myelocytic leukaemia (PML) protein in some of the tested MCC cell lines [128]. PML is one of the IFN-stimulated genes (ISG) that are activated downstream of the IFN signalling pathway and is associated with apoptotic pathways. For the JK virus, another human polyomavirus, it was shown in human glial cells that PML traps free LTA in nuclear bodies and therefore down-regulates it, leading to a reduced infection. Gasparovic et al. assume that IFNβ increases the quantity of PML [129]. If the LTA is trapped inside these PML bodies, it slows down the proliferation of the virus [130].

In addition, class I and II IFNs do not only induce apoptosis of MCC cells but also cause an up-regulation of MHC molecules on the surface of the tumour cells [42]. Another study in MCC cell lines showed that IFNα can inhibit proliferation and induce apoptosis in those tumour cells. Apoptosis was detected by caspase-3 and apoptotic DNA strand breaks [131].

While IFNα and IFNβ show anti-tumour effects, IFNγ is a double-edged sword in cancer. On the one hand, this cytokine is produced by activated T and NK cells leading to MHC up-regulation and helps with the recruitment of new T-cells into the tumour site [132]. On the other hand, it can act immune suppressively, activating the PD-1 axis and leading to an up-regulation of PD-L1 and PD-L2 on cells in the TME, as well as inducing indoleamine-2,3-dioxygenase (IDO) [133]. IDO leads to the depletion of tryptophan, suppressing effector T-cells and activating T_regs_ through the kynurenine pathway. Ayers et al. show that IDO, LAG3, and TIGIT are up-regulated in melanoma through IFNγ and suggest that these tumours have an increased response towards anti-PD-1 therapy [134]. MCC patients with a low mean IDO expression in the tumour cells and low tryptophan-2,3-dioxygenase 2 (TDO2) in the TME showed better survival compared to patients with high IDO and TDO2 [135].

Furthermore, cancer cells often harbour increased amounts of damaged DNA, which activates the protein stimulator of interferon genes (STING). In MCC cell lines, STING expression is repressed, and Liu et al. showed the relevance of STING-activity in the following model system: They transfected MCC cell lines with mutant STING^S162A/G200/Q2661^ which is highly responsive to the synthetic agonist DMXAA. Treatment with DMXAA resulted in a stop of tumour proliferation and reactivated their anti-tumour inflammatory cytokine/chemokine production. This treatment increases the T-cell migration towards the tumour cells and their killing [136]. However, in murine models, a systemic activation of STING resulted in anti-proliferative effects and cell death of T-cells [137]. This method should therefore be considered locally and could be used in combination with intratumoural injections.

### 3.5. MCC T-Cell Epitopes

As described above, most MCC patients bear tumours that have the MCPyV integrated and express the T antigens (i.e., the small TA and a truncated form of the large TA). Hence, from an immunological point of view, MCC is a very special tumour, since it expresses a viral antigen as oncogenic driver. The immune response directed against the T antigens could be especially effective since these antigens are foreign to the individual and no central immune tolerance needs to be overcome. Due to the polymorphism of the MHC genes, the truncLT-epitopes presented on the HLA-A, HLA-B, or HLA-C can vary across patients. Different splicing of the T antigens can also contribute to this variance. Identification of these T-cell epitopes and the associated HLA molecules is crucial for the development of therapeutic vaccines. A vaccine covering different MCPyV-specific T-cell epitopes and HLA types should be used, since monotherapy with adoptive T-cell transfer of CD8^+^ T-cells restricted to one HLA molecule showed tumour evasion by down-regulation of that HLA molecule due to selection pressure [138]. So far, a higher number of specific epitopes could be identified for CD8^+^ [91,104,139,140] than for CD4^+^ T-cells [141,142], but since detection of the latter is technically more challenging and lower number of antigen-specific CD4^+^ T-cells are present in the peripheral blood this may not fully represent reality [141].

Analysis of different MCC samples showed that the T-cell responses were directed against both the LTA and the STA [90]. These T-cells were not found in healthy individuals but were inducible in half of the patients. However, responses against the capsid protein VP1 were existent in both healthy individuals and patients [91]. Iyer et al. identified the LT_92-101_ CD8^+^ T-cell epitope restricted to the HLA-A*24:02 molecule [104]. Lyngaa et al. showed that the presence of HLA-A2* LTA_15-24_ and HLA-A2* STA_171-182_ among specific CD8^+^ T-cells indicates that LTA and STA peptides are processed and presented by the MHC machinery [91].

Samimi et al. suggests that the region between the amino acids 74 and 103 of the LTA is highly immunogenic and they additionally identified two new epitopes recognised by MCPyV-specific T-cells. These epitopes are LT_95-103_ presented in HLA-A*1101, and LT_77-85_ presented in HLA-B*1801. For MCPyV^+^ tumours they propose a vaccine with a synthetic long peptide of 29 amino acids covering the immunogenic region [140]. Jing et al. confirmed the identified epitopes by Samimi et al. in another study and found a slightly extended LT region between amino acids 70–110 to be especially immunogenic [139].

Longino et al. identified a CD4^+^ T-cell epitope (LT_209-228_), which is located within the conserved Rb-binding domain, and could be presented in the context of three different HLA II alleles. In 78% of the MCC patients, they detected specific CD4^+^ T-cells, which were highly enriched in the tumours. The induction or promotion of T-cells specific for this epitope may be a therapeutic option. A modification of the S220 amino acid would still allow recognition by CD4^+^ T-cells of the epitope, but would hamper the LTA from binding to the Rb protein. This detoxification would make it possible to use this epitope in a therapeutic vaccine setting circumventing a potential tumourgenic effect [141].

### 3.6. T-Cells as Potential Predictive Biomarkers for Response to ICI Treatment in MCC

Clinical efficacy of ICIs relies on the reactivation capacity of T-cells. So far, no marker has been found that is able to sufficiently predict the success of the treatment and that allows a differentiation between responders and non-responders before treatment. However, RNA sequencing identified certain gene sets that are up-regulated in non-responders and that could help with the identification.

The expression of PD-1 on the surface of T-cells seems to influence the reactivation efficacy of ICI treatment. Simon et al. showed that co-expression of both PD-1 and TIGIT on CD8^+^ T-cells was associated with better response to ICI treatment. Those double positive CD8^+^ T-cells in the blood of MCC patients seem to define a subpopulation, which should be monitored to assess clinical efficacy of anti-PD-1 antibodies. Additionally, they found an enrichment of HLA-DR^+^ CD38^+^ CXCR5^+^ cells within this DP cell population, whereby CXCR5 represents a marker for CD8^+^ cytotoxic follicular T-cells [143].

A study by Nakamura et al. showed that MCC can be classified into two subtypes via their transcriptome. One type displays an immune-active phenotype with a TCR-related signature, while the other shows a profile related to cell division. The former shows higher PD-L1 expression and better survival, while the latter is highly correlated with the expression of glucose-6-phosphate-dehydrogenase (G6PD). The authors show its value as prognostic marker, since it correlates with tumour activity. Since the expression of G6PD negatively correlated with PD-L1 expression, the authors suggest that a low level of G6PD may act as predictive marker for ICI treatment [144]. G6PD is part of the pentose phosphate pathway (PPP) producing nucleotides and protecting against oxidative cell damage. In other cancer types it could be shown that it works as a prognostic marker [145,146]. High G6PD expression is associated with low activity of the immune system. In contrast, a low G6PD expression is associated with higher activity of the immune system due increased cell death.

Kacew et al. found that single nucleotide variants (SSNV) in AT-rich interactive domain-containing protein 2 (ARID2) and neurotrophic receptor tyrosine kinase 1 (NTRK1) genes may correlate with ICI response. ARID2 is involved in chromatin remodelling, and its loss is associated with increased sensitivity to IFNγ and T-cell-mediated killing. NTRK1 codes for tropomyosin receptor kinase (TrkA) of which the influence on T-cell activity is unclear. The data suggest that the inhibition of ARID2 in ARID2-non-mutated tumours could enhance their ICI response, but this still needs to be evaluated [147].

Weppler et al. identified clinical and imaging factors that could be helpful for prognostic purposes. The factors they associated with increased immune responses to ICI treatment were lack of earlier chemotherapy before ICI treatment, a younger age of the patients at diagnosis, lower baseline [^18^F]-fluoro-2-deoxyglucose PET/CT (FDG-PET/CT) metabolic tumour volume, as well as development of tumour-associated immune-related adverse events (irAEs). They suggest that MCPyV-negative MCC tumours with a higher TMB may be more responsive, although they could not detect a correlation between the response to ICI treatment and the TMB [148].

One last promising computational method to predict cancer immunotherapy response is termed TIDE (Tumour Immune Dysfunction and Exclusion), developed by Jiang et al., which combines the models of the primary tumour immune evasion: T-cell dysfunction in tumours with high TIL rate, and T-cell exclusion in tumours with low TIL rate. Additionally, it predicted new ICI resistance genes such as SERPINB9, whose inhibition might increase the efficiency of the ICI treatment. So far, it is only tested in melanoma but could be used for MCC as well [149].

The above-presented data shows that the knowledge about the role of T-cells in MCC has grown over recent years and a lot of promising markers for the prediction of successful ICI treatment have been found. Nevertheless, further research needs to be done to find reliable markers and durable treatment.

## 4. Conclusions

The purpose of this review was to give an insight into the current state of research for MCC treatment. In recent years, the understanding of MCC pathogenesis, underlying mechanisms, and the immunological background has grown. Although the development of ICIs has improved the outcome of many naïve or pretreated MCC patients, almost 50% do not benefit from this development due to resistances. However, there is a broad diversity of promising immunotherapies currently evaluated in clinical trials, for example the combination of therapeutic DC vaccination or immunostimulants with ICIs, which can improve the outcome in patients resistant to anti-PD-1/-PD-L1 monotherapy. Furthermore, T-cells play a major role in defeating MCC, but often display a reversibly exhausted phenotype in the TME. Especially cytotoxic CD8^+^ T-cells are crucial for the anti-tumour response suggesting their reactivation is very important. Nevertheless, to become activated, they need APCs, which often display down-regulated MHC expression in MCC. Therefore, therapeutic approaches such as the up-regulation of MHC complexes, or the use of pro-inflammatory cytokines such as IL-12 could increase the anti-tumour response together with ICI treatment. The important therapeutic goal is to induce inflammation within MCC nodules and fight off the cancer with the help of the immune system, without inducing a systemic shock.

Collectively the reviewed papers show a broad range of research to find a durable treatment option for mMCC patients who do not respond to ICI treatment. Further studies should investigate treatment options for immune-compromised patients as well, as they have an increased risk of developing MCC and represent approximately 10% of the patients. Additionally, it remains important to find reliable markers that are associated with a response towards ICI treatment. In addition, it would be interesting to see if other immune mechanisms, such as alternative inhibitory receptors, or other immune metabolism pathways in immune cells play a role in the TME, and if they do, how they can be beneficially altered. In conclusion, the diversity of therapeutic alternatives seems to have improved the survival of MCC patients, but further clinical trials are needed to assess the efficacy and durability. Because MCC is a rare cancer, cohorts investigated in clinical trials are often small, therefore patients should be encouraged to participate in clinical trials to facilitate significant results.

## Figures and Tables

**Figure 1 ijms-22-08679-f001:**
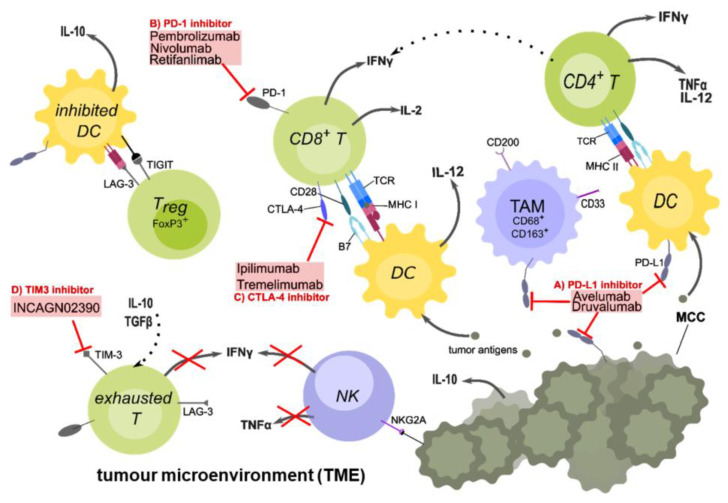
The tumour microenvironment and targets of checkpoint inhibitor therapy in MCC patients. Tumour-cell death causes the release of tumour antigens, which are taken up by antigen-presenting cells (APCs) such as dendritic cells (DCs), to be processed and presented on major histocompatibility complexes II (MHC II) to CD4^+^ or on MHC I to CD8^+^ T-cells. A crucial co-stimulatory signal is given by the interaction between CD28 on the T-cell and the B7 family proteins on the DCs. Upon activation, CD4^+^ T-cells secrete IL-2 and TNFα, which stimulates CD8^+^ T-cell activation and proliferation. DCs secrete IL-12, which is important for the Th1 response. After activation, CD8^+^ T-cells secrete pro-inflammatory cytokines such as IL-2 and IFNγ, and may be able to directly kill tumour cells. However, because of the immunosuppressive TME, inhibitory checkpoint receptors such as PD-1 on the T-cell and its ligand PD-L1 on APCs are up-regulated. Other checkpoints such as CTLA-4, which competes with CD28 for binding B7 are further up-regulated. Regulatory T-cells (Tregs) inhibit DCs via TIGIT and LAG3 binding to MHC II, preventing DCs to activate T-cells. They also secret IL-10, an anti-inflammatory cytokine, which further down-regulates the immune response. Due to chronic antigen exposure, T-cells often display a reversible exhausted phenotype with high expression of inhibitory checkpoint molecules and can therefore be targeted with checkpoint inhibitors (ICIs). There are anti-PD-L1 antibodies (A) such as Avelumab or anti-PD-1 (B) antibodies such as Pembrolizumab and Nivolumab, preventing the interaction of this inhibitory checkpoint receptor with its ligand. Anti-CTLA-4 (C) antibodies such as Ipilimumab are only used in combination with other ICIs in MCC. In currently ongoing clinical trials, other checkpoint inhibitors such as INCAGN02390 (D) to target TIM-3, Druvalumab (A), or Retifanlimab (B) are investigated. Additional cell types found in the TME are natural killer (NK) cells, which express additional inhibitory receptors on their surface (NKG2A) whereby their secretion of pro-inflammatory cytokines such as IFNγ is suppressed. Tumour-associated macrophages (TAMs) are immunosuppressive M2 macrophages which express high levels of PD-L1 and CD200 (OX-2), and prevent the development of inflammatory macrophages. In conclusion, the TME is an immunosuppressive environment that favours tumour growth and evasion rather than killing of the tumour by the immune system. Created with Inkscape.

**Table 1 ijms-22-08679-t001:** Ongoing clinical trials for the treatment of MCC either as monotherapies or in combined treatment with ICIs, Source: Clinicalotrial.gov, July 2021.

Study Treatment	Phase	NCT Number	MCC Status	Study Status	Notes
**PD-1/L1 inhibitors, monotherapy**					
Pembrolizumab	III	NCT03783078	mMCC	active, not recruiting	single-arm, open-label trial
Pembrolizumab	II	NCT02267603	aMCC, stage III A and B, IV	active, not recruiting	first-line treatment, open-label trial
Avelumab (JAVELIN Merkel 200 trial)	II	NCT02155647	mMCC	active, not recruiting	multicentre, international, single-arm, open-label trial
Adjuvant Avelumab (ADAM trial)	III	NCT03271372	MCC Stage III A and B	recruiting	patients who already underwent surgery and/or radiation therapy
Adjuvant Nivolumab or Ipilimumab (ADMEC-O trial)	II	NCT02196961	resected MCC	active, not recruiting	national, open-label, randomised trial
Nivolumab (CheckMate358 trial)	I/II	NCT02488759	MCC and others	active, not recruiting	non-comparative, open-label trial with multiple cohorts
**Other checkpoint inhibitors, monotherapy**					
Retifanlimab (anti-PD-1, POD1UM-201 trial)	II	NCT03599713	aMCC or mMCC	recruiting	single-arm, open-label trial
INCAGN02390 (anti-TIM3 antibody)	I	NCT03652077	MCC and others	active, not recruiting	open-label, dose-escalation, safety, tolerability trial
In-situ vaccination with Tremelimumab (anti-CTLA-4) and IV Druvalumab (anti-PD-L1) + PolyICLC (TLR3 agonist)	I/II	NCT02643303	MCC	recruiting	open-label, multicentre trial of an in-situ vaccine for MCC
CK-301 (anti-PD-L1)	I	NCT03212404	MCC and others	recruiting	open-label, multicentre, dose-escalation study
**PD-1/L1 inhibitor with/without CTLA-4 inhibitors and/or radiation therapy, combination therapy**					
Pembrolizumab with or without Stereotactic Body Radiation Therapy	II	NCT03304639	aMCC, mMCC	recruiting	randomised study of combination therapy, open-label trial
Avelumab with 177-Lu-DOTATATE (type of radiation) (GoTHAM trial)	I/II	NCT04261855	mMCC	recruiting	signal-seeking, biomarker study to evaluate safety of combination therapy
Nivolumab and Ipilimumab ± stereotactic body radiation therapy (SBRT)	II	NCT03071406	mMCC	recruiting	randomised, multi-institutional, open-label trial
Nivolumab + Radiation or Nivolumab + Ipilimumab	I	NCT03798639	MCC stage IIIA, IIIB	recruiting	randomised, multi-institutional pilot study
SO-C1010 ± Pembrolizumab	I/Ib	NCT04234113	aMCC, mMCC	recruiting	multicentre, open-label trial
Avelumab and/or radiation therapy	II	NCT04792073	aMCC	recruiting	prospective, open-label, single-centre
Palliative RT and Anti-PD-1/PD-L1	II	NCT03988647	mMCC	active, not recruiting	open-label study
Avelumab with surgery/radiation (I-MAT)	II	NCT04291885	MCC	recruiting	randomised, placebo-controlled study
Focused Ultrasound Ablation (FUSA) with/without PD-1 with/without Imiquimod	I	NCT04116320	MCC and others	recruiting	pilot evaluation, open-label study
Laser Interstitial ThermoTherapy (LITT) + Pembrolizumab	I	NCT04187872	MCC and others	recruiting	open-label, controlled pilot study
NBTXR3 + radiotherapy + anti-PD-1 antibodies	I	NCT03589339	MCC and others	recruiting	prospective study
**PD-1/L1 inhibitor with TLR agonists, combination therapy**					
NKTR-262 (TLR agonist) + NKTR214 (Bempegaldesleukin, IL-2 agonist) or NKTR214 + Nivolumab (REVEAL trial)	I/II	NCT03435640	locally aMCC, mMCC	recruiting	open-label, multicentre, dose escalation and dose-expansion study
Pembrolizumab + Cavrotolimod (TLR9 Agonist) or Cemiplimab (anti-PD-1) + Cavrotolimod	Ib/II	NCT03684785	aMCC, mMCC and others	recruiting	open-label, two-part, multicentre trial
**PD-1/L1 inhibitor combined with interleukin agonists or other therapies, combination therapy**					
Avelumab + N-803 (IL-15 agonist) + haNK (QUILT-3.063 trial)	II	NCT03853317	MCC	recruiting	relapsed after ICI treatment, single-arm trial
Atezolizumab (anti-PD-L1) + NT-I7 (recombinant human IL-7)	I/II	NCT03901573	MCC	recruiting	naïve or anti-PD-1/PD-L1 relapsed MCC, test if addition of NT-I7 provides clinical benefit
Atezolizumab + Bevacizumab (VEGF inhibitor)	II	NCT03074513	MCC and others	active, not recruiting	single-arm, open-label trial
Tacrolimus (calcineurin inhibitor), Nivolumab and Ipilimumab	I	NCT03816332	mMCC, Stage III A and B	recruiting	kidney transplant recipients, open-label trial
Pembrolizumab ± XmAb18087 (bispecific antibody)	I	NCT04590781	recurrent mMCC	not yet recruiting	multiple dose study to evaluate safety of XmAb18087
Avelumab + Dominostat (HDAC inhibitor, MERKLIN2 trial)	II	NCT04393753	aMCC	recruiting	relapsed after ICI treatment, single-arm trial
Avelumab + gene modified immune cells (FH-MCVA2TCR)	I/II	NCT03747484	mMCC	recruiting	relapsed after ICI treatment, single-arm trial
Zimberelimab (anti-PD-1 antibody) + Etrumadenant (adenosine receptor antagonist)	I	NCT03629756	MCC and others	active, not recruiting	open-label, dose-escalation, dose-expansion study
BT-001 (oncolytic virus) with/without Pembrolizumab	I/II	NCT04725331	MCC	recruiting	multicentre, open-label, consecutive cohorts, dose-escalation study
OC-001 with/without anti-PD-1 or anti-PD-L1 antibodies	II/II	NCT04260802	MCC and others	recruiting	two-part, open-label, multicentre study
Nivolumab + talimogene Laherparepvec (modified oncolytic herpes virus)	II	NCT02978625	Non-melanoma skin cancers	recruiting	open-label study
Plinabulin (targets new blood vessels) + Avelumab/Atezolizumab/Durvalumab/Nivolumab/Pembrolizumab/radiation therapy	I/II	NCT04902040	MCC and others	recruiting	open-label, single-centre study, after progression of PD-1 or PD-L1 targeted antibodies
Neo-adjuvant Lenvatinib (multikinase inhibitor) + Pembrolizumab	II	NCT04869137	MCC	recruiting	open-label study
Avelumab + Domatinostat (MERKLIN 1)	II	NCT04874831	mMCC	not yet recruiting	treatment-naïve mMCC, multicentre, open-label trial
N-803 + Pembrolizumab/Nivolumab/Atezolizumab/Avelumab/Durvalumab/Pembrolizumab with or without PD-L1 t-haNK (QUILT-3.055)	II	NCT03228667	MCC and others	active, not recruiting	multicohort, open-label trial
**Adoptive cell transfer with/without ICIs, with/without radiation, combination therapy**					
Transfer of allogenic BK specific cytotoxic T lymphocytes	II	NCT02479698	cancers with BK or JC virus, MCC and others	recruiting	open-label trial
Localized radiation therapy or recombinant IFNβ and Avelumab in combination with or without adoptive immunotherapy	I/II	NCT02584829	mMCC, Stage IV	active, not recruiting	open-label trial, cellular adoptive immunotherapy + ICI treatment
iPS cell derived cells (NK cells, FT500) + one of the ICIs (Nivolumab, Pembrolizumab, Atezolizumab) (FT500-101)	I	NCT03841110	MCC and others	recruiting	open-label trial, FT500 = off-the-shelf, iPSC-derived NK cells
FT500 (allogenic NK cells)—long-term follow-up of FT500-101		NCT04106167	MCC and others	recruiting	multicentre, non-interventional, observation study
Fludarabine + Cyclophosphamide + tumour-infiltrating lymphocytes (TILs)	II	NCT03935893	MCC and others	recruiting	open-label study
**Additional immunotherapeutic options**					
Immunotherapy with KRT-232 (inhibitor of MDM2)	II	NCT03787602	MCC	recruiting	relapsed after ICI treatment, open-label trial
Immunotherapy with Ifx-Hu2.0 vaccine	I	NCT04160065	aMCC	recruiting	intralesional immunotherapy
aNK infusions in combination with ALT-803 (IL-15 agonist, QUILT-3.009 trial)	II	NCT02465957	MCC, stage III or IV	active, not recruiting	multicentre, non-randomised, open-label trial determining effects of aNK in ALT-803 combination
T-VEC (oncolytic virus) ± hypo fractioned radiotherapy	II	NCT02819843	MCC	recruiting	randomised trial of intralesional talimogene laherparepvec (oncolytic virus)
Immunotherapy with TAEK-VAC-HerBy vaccine	I/II	NCT04246671	MCC and other (HER2 expressing cancers)	recruiting	open-label, expansion cohorts trial of intravenous vaccine administration
T-VEC (20139157 T-VEC)	I	NCT03458117	MCC and others	recruiting	open-label trial
Immunotherapy with the small molecule INCB099318	I	NCT04272034	MCC	recruiting	2 parts, part 1: dose escalation, part 2: explore safety, effects, pharmacokinetics
Immunotherapy with the small molecule INCB099280	I	NCT04242199	MCC	recruiting	2 parts, part 1: dose escalation, part 2: explore safety, effects, pharmacokinetics
Cabozanitinib (XL184, multiple receptor tyrosine kinase inhibitor)	II	NCT02036476	mMCC	active, not recruiting	Open-label, non-randomised, patients progressed after platinum-based therapy

ICI: immune checkpoint inhibitor, MCC: Merkel cell carcinoma, mMCC: metastatic Merkel cell carcinoma, aMCC: advanced Merkel cell carcinoma, BK: human polyomavirus 1, JC: human polyomavirus 2, aNK: activated natural killer cells.

**Table 2 ijms-22-08679-t002:** Results of clinical trials evaluating immune checkpoint inhibitors as a treatment option for MCC.

Drug	Reference	Phase	n	Line of Treatment	Dose	ORR	PFS	OS
**Avelumab**	Kaufmann et al., 2016 [62]	II	35	second-line treatment	10 mg/kg	31.8%	NR	NR
	D’Angelo et al., 2018 [65]	II	39	first-line treatment	10 mg/kg	62.1%	3.1–4.6 months	NR
	D’Angelo et al., 2020 [64]	II	88	second-line treatment	10 mg/kg	33%	24% (24 months)	12.6 months
**Pembrolizumab**	Nghiem et al., 2016 [59]	II	26	first-line treatment	2 mg/kg	56%	67% (6 months)	NR
	Nghiem et al., 2019 [60]	II	50	first-line treatment	2 mg/kg	56%	48.3% (24 months)	68.7% (24 months)
**Nivolumab**	Topalian et al., 2020 [74]	I/II	39	neo-adjuvant	240 mg	pCR: 47.6%	NR	NR

n: number of participants, NR: not reported, ORR: overall response rate, PFS: median progression free survival rate, OS: overall survival, pCR: pathological complete response.

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
