# Peer review of "T-Cell Responses in Merkel Cell Carcinoma: Implications for Improved Immune Checkpoint Blockade and Other Therapeutic Options"

_ijms, 2021, doi:10.3390/ijms22168679_

Round 1

Reviewer 1 Report

This is an extensive review article dealing with immunological aspects of Merkel cell carcinoma relevant for modern therapeutic approaches. Although being rather voluminous it comprises all aspects of present and future immunotherapy in this particular rare but highly malignant skin tumor. It has been carefully written and profoundly discusses the different immunological aspects and possible therapeutic options of this tumor.

There is one point which should be considered by the authors:

The review includes a large summarizing table containing 52 ongoing trials (Table 2) which is only briefly referred to in the text (lines 464-468). Here, a better link between this table and the text would be appropriate, i. e. specific references to this table in all relevant text parts. For example, the authors refer in lines 371-379 to one particular study among these ongoing studies (NCT04393753) but in this section do not refer to Table 2 (which would seem to be required). Furthermore, the text already states results of this study which according to Table 2 is still recruiting. It should be clarified why there are already results available. Thus, the text in general should refer to the relevant ongoing studies in each section where this is appropriate.

Author Response

First of all, we thank the reviewer for reading and evaluating our manuscript.

To the raised concerns, we have prepared a point-by-point response below:

This is an extensive review article dealing with immunological aspects of Merkel cell carcinoma relevant for modern therapeutic approaches. Although being rather voluminous it comprises all aspects of present and future immunotherapy in this particular rare but highly malignant skin tumor. It has been carefully written and profoundly discusses the different immunological aspects and possible therapeutic options of this tumor.

There is one point which should be considered by the authors:

The review includes a large summarizing table containing 52 ongoing trials (Table 2) which is only briefly referred to in the text (lines 464-468). Here, a better link between this table and the text would be appropriate, i. e. specific references to this table in all relevant text parts. For example, the authors refer in lines 371-379 to one particular study among these ongoing studies (NCT04393753) but in this section do not refer to Table 2 (which would seem to be required). Furthermore, the text already states results of this study which according to Table 2 is still recruiting. It should be clarified why there are already results available. Thus, the text in general should refer to the relevant ongoing studies in each section where this is appropriate.

Answer: We improved the link between the table and the text by adding specific references to this table in all relevant text parts, always naming the NCT-number (lines 216, 283, 301, 310, 320, 374, 463, and 499). We also rephrased the text at several sites to better differentiate between clinical trials, case reports, and observational studies to better indicate why some are referred in the table, and others are not. Due to the earlier reference to table 2, we changed the order of the tables.

Concerning the results of the one particular study (NCT04393753) you are referring to: This study is still recruiting and no results are available yet. The sentence afterwards was unintentionally written in an ambiguous way as it was meant to refer not to NCT04393753 but to the study by Song et al., 2020. We clarified the sentence, so that it is now clear that the results discussed in this section do not refer to NCT04393753. (lines 454 to 461)

Reviewer 2 Report

Major points.

1/ PDL-1 negative MCC is associated with worse survival? Discussed why therefore use of anti-PDL1 is efficient as first line of treatment?

2/ Ectopic lymphoid organ should be discussed somewhere in this review and how ICI therapy may interfere with those structure, the quality of anti-MCPyV T antibodies and the patient response. In fact, this ectopic lymphoid are associated with overall and recurrence free survival in virus induced or positive MCC (PMID: 25550797: Int J Clin Exp Pathol, 2014 Oct 15;7(11):7610-21.eCollection 2014: “Prognostic value of immune cell infiltration, tertiary lymphoid structures and PD-L1 expression in Merkel cell carcinomas”. HIV infection is known to negatively interfere with germinal center B cells and Tfh CXCR5hi or + (T follicular helper cells [that are Tcm]) and by extend to Ab quality via PD-1/PDL-1 interference. Tfh-like function could also be discussed. Thus, ectopic lymphoid organ (CXCR5+) may be with better response to ICI therapy by the local presence of CD8+ CXCR5+ cytotoxic CD8 and better Ab response.

3/ Lines 549 or Line 612: TIGIT (-) and CD226 (+) that both to PVR (CD155) are both induced by IL-15 on NK cells. Therefore IL-15 + anti-TIGIT has been highlithed as a therapeutic option for refractory melanoma (https://clincancerres.aacrjournals.org/content/26/20/5520) and would be of high interest to discuss in MCC context.

4/ Vaccination with mRNA coding for MCC neoantigen should be discussed over DC-induced vaccine strategies (in the current sanitary context).

Minor points.

1/Abstract: line 18-19 Pembrolizumab is an anti-PD-1, therefore using “anti-PD-1/PDL-1 – immune checkpoint inhibitors” seems more appropriate than anti-PD(L)1.

2/Abstract: line 20: develop an ICI induced- immune related adverse events (irAEs) would be rather used than anti-PD(L)1 refractory disease.

2/ Line 405: anti-PD1/anti-PD-1 antibodies should be corrected.

3/ Lines 171-172: ‘the Titre of these antibodies and line 172: “the titre” should be replaced by: anti-MCPyV T antibodies titer decreases with successful treatment, while rebound upon recurrence. For these reasons, anti-MCPyV T antibodies titer is used…

4/ Consider changing the title for better accuracy with manuscript content: T-cells responses in Merkel cell carcinoma: implication for improved IC blockade and others therapeutical options?

4/ In general, English rephrasing should be considered to improve the manuscript flow.

Author Response

First of all, we thank the reviewer for reading and evaluating our manuscript.

To the raised concerns we have prepared a point-by-point response below:

Major points.

1/ PDL-1 negative MCC is associated with worse survival? Discussed why therefore use of anti-PDL1 is efficient as first line of treatment?

Answer: We added an additional paragraph in the Avelumab section where we discuss the efficiency of anti-PD-L1 therapy for PD-L1 negative patients (lines 342-348).

In general, PD-L1 expression seems to be associated with improved survival in MCC patients compared to MCC patients lacking PD-L1 expression (Lipso et al., 2013). Nevertheless, Kaufman et al, 2016 showed that patients respond to Avelumab (anti-PD-L1) treatment, independent of PD-L1 or MCPyV status, suggesting different underlying mechanisms cause the therapeutic benefit.

2/ Ectopic lymphoid organ should be discussed somewhere in this review and how ICI therapy may interfere with those structure, the quality of anti-MCPyV T antibodies and the patient response. In fact, this ectopic lymphoid are associated with overall and recurrence free survival in virus induced or positive MCC (PMID: 25550797: Int J Clin Exp Pathol, 2014 Oct 15;7(11):7610-21.eCollection 2014: “Prognostic value of immune cell infiltration, tertiary lymphoid structures and PD-L1 expression in Merkel cell carcinomas”. HIV infection is known to negatively interfere with germinal center B cells and Tfh CXCR5hi or + (T follicular helper cells [that are Tcm]) and by extend to Ab quality via PD-1/PDL-1 interference. Tfh-like function could also be discussed. Thus, ectopic lymphoid organ (CXCR5+) may be with better response to ICI therapy by the local presence of CD8+ CXCR5+ cytotoxic CD8 and better Ab response.

Answer: We included tertiary lymphoid structures (TLS) as a relevant structure of the MCC tumor microenvironment and highlighted their role in immune checkpoint blockade. We also discussed the controversial role of B-cell and antibody-responses, which correlate with tumor mass and are sometimes increased in non-responders, but also indicate broad immune responses against the tumors (lines 728-744).

3/ Lines 549 or Line 612: TIGIT (-) and CD226 (+) that both to PVR (CD155) are both induced by IL-15 on NK cells. Therefore IL-15 + anti-TIGIT has been highlithed as a therapeutic option for refractory melanoma (https://clincancerres.aacrjournals.org/content/26/20/5520) and would be of high interest to discuss in MCC context.

Answer: We included this combined therapeutic option as promising treatment strategy in context of MCC (line 655-659)

4/ Vaccination with mRNA coding for MCC neoantigen should be discussed over DC-induced vaccine strategies (in the current sanitary context).

We included an additional paragraph, that describes direct vaccination with RNA in other tumour entities and suggested vaccination with mRNA of MCC-neoantigens as potential therapeutic vaccination apart from DC vaccines (line 540-552).

Minor points.

1/Abstract: line 18-19 Pembrolizumab is an anti-PD-1, therefore using “anti-PD-1/PDL-1 – immune checkpoint inhibitors” seems more appropriate than anti-PD(L)1.

Answer: We corrected this and changed all PD(L)-1 to PD-1/PDL-1.

2/Abstract: line 20: develop an ICI induced- immune related adverse events (irAEs) would be rather used than anti-PD(L)1 refractory disease.

Answer: We exchanged anti-PD(L)1 refractory disease with ICI induced- immune related adverse events (irAEs)

2/ Line 405: anti-PD1/anti-PD-1 antibodies should be corrected.

Answer: We changed anti-PD1/anti-PD-1 to anti-PD-1/-PD-L1.

3/ Lines 171-172: ‘the Titre of these antibodies and line 172: “the titre” should be replaced by: anti-MCPyV T antibodies titer decreases with successful treatment, while rebound upon recurrence. For these reasons, anti-MCPyV T antibodies titer is used…

Answer: We changed this to: Anti-MCPyV T antibodies titre decreases after successful treatment and upon recurrence they increase again (line 193).

4/ Consider changing the title for better accuracy with manuscript content: T-cells responses in Merkel cell carcinoma: implication for improved IC blockade and others therapeutical options?

Answer: We totally agree and changed the title to ,,T-cell responses in Merkel cell carcinoma: implication for improved immune checkpoint blockade and other therapeutic options”

4/ In general, English rephrasing should be considered to improve the manuscript flow.

Answer: Throughout the whole manuscript, we rephrased English expressions in an appropriate way.